# Temperature-Sensitive Point Selection and Thermal Error Model Adaptive Update Method of CNC Machine Tools

Hui Liu [1,*], Enming Miao [2], Jingfan Wang [3], Liyin Zhang [1] and Siyu Zhao [4]

1 School of Automation, Xi'an University of Posts & Telecommunications, Xi'an 710121, China; zhangliyin@xupt.edu.cn
2 College of Mechanical Engineering, Chongqing University of Technology, Chongqing 400054, China; miaoem@cqut.edu.cn
3 Shaanxi Institute of Metrology Science, Xi'an 710100, China; w_jf0412@126.com
4 Department of Chemical Engineering, University College London, London WC1E 7JE, UK; s.zhao.17@ucl.ac.uk
* Correspondence: liuhui@xupt.edu.cn

**Abstract:** The thermal error of CNC machine tools can be reduced by compensation, where a thermal error model is required to provide compensation values. The thermal error model adaptive update method can correct the thermal error model by supplementing new data, which fundamentally solves the problem of model robustness. Certain problems associated with this method in temperature-sensitive point (TSP) selection and model update algorithms are investigated in this study. It was found that when the TSPs were selected frequently, the selection results may be different, that is, there was a variability problem in TSPs. Further, it was found that the variability of TSPs is mainly due to some problems with the TSP selection method, (1) the conflict between the collinearity among TSPs and the correlation of TSPs with thermal error is ignored, (2) the stability of the correlation is not considered. Then, a stable TSP selection method that can choose more stable TSPs with less variability was proposed. For the model update algorithm, this study proposed a novel regression algorithm which could effectively combine the new data with the old model. It has advantages for a model update, (1) fewer data are needed for the model update, (2) the model accuracy is greatly improved. The effectiveness of the proposed method was verified by 20 batches of thermal error measurement experiments in the real cutting state of the machine tool.

**Keywords:** thermal error; temperature-sensitive points; model update algorithms; adaptive update method

## 1. Introduction

The thermal error of CNC machine tools is the cutting tool offset caused by thermal deformation during the working process. It is the main reason for the deterioration of machine tool accuracy when processing for a long time [1,2]. This problem can be solved by thermal compensation, in which the value of thermal error is used as a negative feedback signal for compensation [3]. The thermal error cannot be measured directly during machining because the tool is rotating at high speed. Therefore, a feasible approach is to predict thermal error by temperature [4]. However, this also brings a new question: How should the relationship between temperature and thermal error (known as thermal error model) be determined? Lo and Ni [5] proposed that thermal error can be predicted only by measuring the temperature of certain points on the machine tools (called temperature-sensitive points (TSPs)). The establishment of the thermal error model consists of two main steps:
(1) Selection of the TSPs.
(2) Building a mathematical model between the TSPs and thermal error.
Yang [6] simulated the thermal deformation of the spindle through FEM and also found that several TSPs are sufficient to predict thermal errors. However, some assumptions

are made in the material properties, geometry and heat transfer characteristics during the simulation: the spindle is made of isotropic material, nuts and some holes in the spindle are ignored, temperature change of the coolant are ignored, etc. Wei [7] found by simulation that TSPs with a linear relationship with thermal error are better for the thermal error model, the spindle is treated as a round bar during simulation. Zhang [8] found that the thermal conductivity of the contact surfaces of the machine tool structures is affected by the way of assembly. Then, combined with experiments, fractal theory and the hybrid genetic algorithm were used for modeling the thermal conductivity, which provided a reference for the design and assembly of the spindle. According to the above studies, the thermal deformation characteristics are affected by the machine tool structure geometry and the way of assembly. However, a machine tool is a complex device composed of multiple parts, and it is very difficult to quantitatively obtain the specific values of the geometric shape and thermal deformation parameters of each structure. The current related research has to make some assumptions, but it also leads to a decrease in the accuracy of the model, and only some qualitative conclusions can be obtained. Therefore, it is challenging to determine the relationship between machine tool temperature and thermal error in a purely theoretical way [9,10]. Therefore, data-driven modeling is a commonly used method. This method measures thermal error and temperatures near major heat sources while the machine remains idle. Subsequently, it selects TSPs from initial temperature measurement points and builds the model based on the measured data [11]. To improve the prediction accuracy of thermal error, there has been much research on TSP selection methods and modeling algorithms in recent years [12]. However, data-driven methods can only fit thermal error properties contained in the measured data. If the thermal error characteristics are changed, the model accuracy decreases. Liu [13] found that the ambient temperature causes the thermal error characteristics to change, which can only be solved by remeasuring the data and building a new model. The institute for Machine Tools and Manufacturing in Switzerland [14–17] also noticed this problem and proposed an adaptive update method for the thermal error model. It consists of the following steps: setting up the thermal error online detection system on the machine tool; periodically halting the machining and measuring the thermal error and temperature; when the difference between the predicted and measured value of thermal error is beyond the tolerance range, the No-Good mode is entered, and the thermal error and temperature are remeasured to build a new model [14,15]. Zimmermann et al. [16] found that the location of TSPs would also change and added a TSP reselection step in the model update process. Zimmermann [17] also studied the timing of the model update. The above-mentioned works focus on the adaptive update method of the thermal error model and research in depth the TSP selection and thermal error modeling.

**Variability of TSP selection results**

Zimmermann [17] found that the TSPs will change under different conditions. Miao [18] carried out several thermal error measurement experiments on a vertical machining center under different conditions and observed that the selection results of TSPs are variable. At present, there is much research content on the TSP selection method. However, only one piece of experimental data is used to select the TSPs, and the consistency of the selection results of TSPs under multiple experiments has not been studied yet. Therefore, it is unknown whether the TSPs have really changed or the selection method is problematic. This is an issue worthy of attention after applying the thermal error model adaptive update method. Because TSPs are frequently selected in the process of model updating, once the wrong TSPs are selected owing to an inappropriate method, the prediction accuracy of the updated model will decrease. This study analyzed the selection method of TSPs from a large batch of thermal error measurement data and found the methodological reasons for the variability problem. The two specific problems are as follows:

**Note: The correlation mentioned in the following text corresponds to the correlation with thermal error, and the collinearity corresponds to the correlation with other temperature measurement points.**

(1) The conflict between correlation and collinearity.

In the current research, the selection methods of TSPs are all based on one strategy: Group selection. First, multiple groups are set up and the initial temperature measurement points with similar changes are placed in the same group. Then, one TSP is selected from each group; the selected TSP must have the highest correlation in the group [19]. The purpose of grouping is to reduce the collinearity of TSPs because the high-collinearity TSPs may increase the probability of model overfitting and reduce robustness [20]. Fuzzy clustering is a commonly used grouping algorithm. This method measures the collinearity by the correlation coefficient. Miao [21], Abdulshahed [22], Zhang [23], Qin [24], and Li [25] all use this method to select TSPs for different types of machine tools. In addition, there are some other methods with the same effect as that of the grouping method. Tsai [26] calculated the collinearity by the condition number of singular value decomposition (SVD) of the temperature measurement value matrix. Fu [27] measured the temperature field of the machine tool through an infrared thermal imager and divided the machine tool into different temperature regions for grouping. Although this reduces collinearity, it also creates a problem [28]. The primary requirement of TSPs is a high correlation with thermal errors. If low collinearity is required at the same time, the high correlation of different TSPs must be intransitive: The TSPs $T_i$, $T_j$ all have high correlations with thermal error, but the collinearity between $T_i$ and $T_j$ is low. The temperature at different positions is inevitably coupled because of heat conduction, and thus it is difficult to meet this requirement [29]. This means that reducing the collinearity and improving the correlation of TSPs is a potential conflict problem. Although the grouping method can select the TSPs with low collinearity, it also makes the correlation low.

(2) The stability of the correlation.

Wei [7] found that good TSPs will show a high linear correlation with thermal error. Current studies often use correlation coefficients to measure the correlation of TSPs [21,23–25,27]. Our recent study [30] found that the correlation of TSPs may be unstable. Errors in the measurement data, changes of machining state, and other factors can change in the correlation. However, this problem is not considered in the current correlation calculation process.

The above two problems result in a low and unstable correlation of TSPs. Because the correlation ranking is the basis for the selection of TSPs, changes in the correlation may lead to changes in TSP selection results, i.e., variability. This study proposed a new method for selecting high-correlation and stable TSPs (hereinafter called "the stability TSPs selection method").

**Adaptive update method of the thermal error model**

If model adaptive updating is not considered, thermal error modeling data are sufficient and modeling speed need not to be considered. After applying the model adaptive update method, the modeling algorithm must be able to learn new information online and use a small amount of data to build a new model quickly and accurately. Blaser [14] and Mayr [15] use multiple linear regression for the model updating. When the prediction error of a model is too large, 12 sets of data are measured and a new model is established. Simplicity and fast computation are the advantages of multiple regression algorithms. However, it cannot remember the old model but can only remodel with new data. This leads to information loss during the model update process. Because machine tool structure and heat source location are important factors influencing thermal error characteristics, this information is constant and implicit in the old model. Hence, the old thermal error model still contains some valid information. Furthermore, how to fuse the old model with new data to improve the performance of the updated model is a problem worthy of study.

Neural network (NNs) and multiple regression are both commonly used for thermal error modeling. For adaptive update modeling algorithms, one advantage of NN is the ability to learn new data based on old models. However, the NN model is difficult to train. It is easy to overfit or underfit, especially for small sample data modeling. In this regard, researchers have conducted much work to find optimal model parameters, such as the ant

colony algorithm [31] and genetic algorithm [32]. Li [33] used the shark smell optimization method to train the NN model, the thermal error of axial and C-axis of five-axis machining was modeled and the fitting accuracy was 3 μm. Huang [34] used a bat-algorithm-based backpropagation NN to model the thermal positioning error of an in-house three-axis experimental platform, and the residual standard deviation of the model was 3.8 μm. The above research improves the accuracy of the model by improving the training method. However, since the loss function of the NN is not convex and cannot be explained, this problem is difficult to completely solve.

The model built by the multiple regression algorithm is more stable and easier to train because of the simple structure and convex loss function. Liu [35] compared the long-term thermal error prediction accuracy of NN and the improved regression algorithm (ridge regression), and proved that the latter is more accurate and stable. Zhang [23] used sliced inverse regression to model the axial thermal error of a horizontal machine tool, and the fitting accuracy was 2.5 μm. Fu [36] used the multiple regression algorithm to model the axial thermal error of the spindle, and the model prediction accuracy was 2 μm. Chen [37] used the multiple regression algorithm to model the Y- and Z-axis thermal error of a vertical machining center, the mean square errors are 3.48 and 4.46 μm, respectively. By comparing the references [33,34] and [23,36,37], it can be inferred that the regression algorithm can achieve the same thermal error modeling accuracy as that of the NN. This is because the relationship between TSPs and thermal error is essentially linear.

In summary, for the model adaptive update method, NN has the advantage of learning online, but it is unstable and difficult to train. The multiple regression cannot learn online, but it is simple, stable, and easy to train. Therefore, this study aimed to improve the multiple regression and endow the regression algorithm with the ability to remember the old model. It was experimentally verified that the proposed algorithm (hereinafter called "update regression") could build a higher precision thermal error model with fewer data.

* In Appendix A, the author's team's research history of thermal error is briefly described, which helps to better understand the intention of this article.

## 2. Typical Thermal Error Modeling Methods

There has been much research on thermal error modeling algorithms, aiming to select the TSPs with low collinearity and high correlation and build the thermal model with high accuracy and robustness. In this section, the typical thermal error modeling algorithms are introduced.

Suppose that there are $M$ initial temperature measurement points, denoted by $T_1, \ldots, T_M$, and the thermal error is denoted by $E$. During the measurement, $N$ times of measurement data are obtained, and are denoted by,

$$\begin{cases} \boldsymbol{T_1} = (t_{1,1}, \ldots, t_{1,N})^{\mathrm{T}} \\ \quad\quad\vdots \\ \boldsymbol{T_M} = (t_{M,1}, \ldots, t_{M,N})^{\mathrm{T}} \\ \quad \boldsymbol{E} = (e_1, \ldots, e_N)^{\mathrm{T}} \end{cases} \tag{1}$$

where $\boldsymbol{T_1}, \ldots, \boldsymbol{T_M}$ are the vectors of the measured temperature values, and $\boldsymbol{E}$ is the vector of the measured thermal error. The symbols in Equation (1) are used in subsequent equations.

### 2.1. TSP Selection Method

The initial temperature measurement points are divided into different groups. The collinearity of the same group is high, whereas the collinearity of different groups is low. The fuzzy clustering is a commonly used algorithm.

(1) Build a fuzzy similarity matrix $\boldsymbol{R} = \left[r_{i,j}\right]_{M \times M}$, where $r_{i,j}$ is the correlation coefficient of $T_i$ and $T_j$.



(2) Convert similarity matrix to equivalent matrix. Perform multiple square fuzzy operations on fuzzy similarity matrices as follows:

$$\left.\begin{array}{c} \boldsymbol{R} \times \boldsymbol{R} = \boldsymbol{R}^2 \\ \boldsymbol{R}^2 \times \boldsymbol{R}^2 = \boldsymbol{R}^4 \\ \vdots \\ \boldsymbol{R}^{2^v} \times \boldsymbol{R}^{2^v} = \boldsymbol{R}^{2^{(v+1)}} \end{array}\right\} \tag{2}$$

The square fuzzy operation is as follows.

$$\boldsymbol{R} \times \boldsymbol{R} = \left[ r_{i,j}{}^2 \right]_{M \times M} \tag{3}$$

$$r_{i,j}{}^2 = \max_{k=1,\dots,M} \left( \min\left( r_{i,k}, r_{k,j} \right) \right) \tag{4}$$

If $\boldsymbol{R}^{2^v} = \boldsymbol{R}^{2^{(v+1)}}$, then $\boldsymbol{R}^{2^v}$ is a fuzzy equivalent matrix. The purpose of transforming the fuzzy equivalent matrix is to make the fuzzy relationship transitive (if $F(T_i, T_k) > \Lambda$ and $F(T_j, T_k) > \Lambda$, then $F(T_i, T_j) > \Lambda$, $\forall\, i, j, k$, $F(*, *)$ is a fuzzy relationship).

(3) Set $\Lambda$ as the threshold. If $T_i$ and $T_j$ are in the same group, then $F(T_i, T_j) > \Lambda$. The number of groups and the number of TSPs are determined $\Lambda$.

The correlations of all temperature measurement points in each group are sorted from large to small, and the largest is TSP. The correlation is calculated by the correlation coefficient as follows:

$$\rho_{T_i,E} = \frac{\sum_{k=1}^{n} \left( t_{i,k} - \overline{t_i} \right) \left( e_k - \overline{e} \right)}{\sqrt{\sum_{k=1}^{n} \left( t_{i,k} - \overline{t_i} \right)^2 \sum_{k=1}^{n} \left( e_k - \overline{e} \right)^2}} \tag{5}$$

where $\overline{t_i}$ and $\overline{e}$ are the averages of the data $\boldsymbol{T}_i$ and $\boldsymbol{E}$, respectively.

*2.2. Thermal Error Modeling Algorithms*

The thermal error model is built from measurement data. The TSPs are the input of the model, and the thermal error is the output. NN and multiple regression are the most frequently used algorithms. The structures and training methods of these two algorithms are different. An NN consists of multiple layers of nodes. The input is nonlinearly processed at each layer, and the processing result of the last layer is the output. The NN is trained by continuously adjusting the connection weights between nodes, and a commonly used method is the steepest descent method. Since the regression algorithm is the main research object of this paper, the NN will not be introduced in more detail for reasons of space.

The model structure of the multiple regression algorithm is polynomial. Here, two regression algorithms are introduced: the ridge regression and ordinary multiple regression. For the ordinary multiple regression, the high collinearity of the model input will increase the estimated variance of model coefficients, making the model unstable and prone to overfitting. Ridge regression adds a regularization term to the loss function, which can significantly reduce variance when there is high collinearity input. The model is expressed as follows:

$$\hat{E} = \beta_0 + \beta_1 T_1 + \cdots + \beta_S T_S \tag{6}$$

where $T_1$–$T_S$ are TSPs, $S$ is the number of TSPs and $\hat{E}$ is the thermal error. Through ridge regression, the model coefficient $\boldsymbol{\beta} = (\beta_0, \beta_1, \dots, \beta_S)^{\mathrm{T}}$ can be estimated by the following equation.

$$\boldsymbol{\beta} = \left( \boldsymbol{A}^T \boldsymbol{A} + p_{ridge} \boldsymbol{I} \right)^{-1} \boldsymbol{A}^T \boldsymbol{E} \tag{7}$$

$$A = \begin{pmatrix} 1 & t_{1,1} & \cdots & t_{S,1} \\ \vdots & \vdots & \ddots & \vdots \\ 1 & t_{1,N} & \cdots & t_{S,N} \end{pmatrix} \tag{8}$$

where *I* is the identity matrix, and $p_{ridge}$ is the ridge parameter. According to a previous study [35], the value of $p_{ridge}$ is suitable between 10 and 30; this study took 25.

When $p_{ridge}$ is zero, it is an ordinary multiple regression algorithm.

## 3. TSP Selection and Thermal Error Model Update Method

### 3.1. Stability TSP Selection Method

For TSPs, low collinearity and high correlation are conflicting states. In particular, the grouping algorithm reduces both the collinearity and the correlation of TSPs. Considering that the ridge regression algorithm can solve the collinearity problem [38], the correlation can be considered as the preferred basis for the selection of TSPs. In addition, a high correlation does not mean that the correlation is stable. The instability of the correlation means that the connection between TSPs and thermal errors is easily disturbed. Therefore, adding a method for evaluating the stability in the correlation calculation is beneficial to improve the quality of TSPs. In this study, the uncertainty was used to determine the stability of correlation; therefore, when selecting TSPs, the correlation and stability were considered comprehensively. This method is called "the uncertainty-correlation coefficient calculation method." The details are as follows.

The correlation coefficient between $T_i$ and $E$ is shown in Equation (5). The uncertainty of the correlation coefficient is as follows.

$$U_{\rho_{T_i,E}} = \sqrt{ U_E^2 \sum_{k=1}^{N} \left( \frac{\partial \rho_{T_i,E}}{\partial e_k} \right)^2 + U_{T_i}^2 \sum_{k=1}^{N} \left( \frac{\partial \rho_{T_i,E}}{\partial t_{i,k}} \right)^2 } \tag{9}$$

$$\frac{\partial \rho_{T_i,E}}{\partial t_{i,k}} = \left(1 - \frac{1}{N}\right) \cdot \frac{(e_k - \bar{e})\sqrt{\sum_{j=1,j\neq k}^{N}\left(t_{i,j}-\bar{t_i}\right)^2 \sum_{j=1}^{N}\left(e_j-\bar{e}\right)^2} - \frac{\left(t_{i,k}-\bar{t_i}\right)\sum_{j=1}^{N}\left(\left(t_{i,j}-\bar{t_i}\right)\left(e_j-\bar{e}\right)\right)\sqrt{\sum_{j=1}^{N}\left(e_j-\bar{e}\right)^2}}{\sqrt{\sum_{j=1}^{N}\left(t_{i,j}-\bar{t_i}\right)^2}}}{\sum_{j=1}^{N}\left(t_{i,j}-\bar{t_i}\right)^2 \sum_{j=1}^{N}\left(e_j-\bar{e}\right)^2} \tag{10}$$

$$\frac{\partial \rho_{T_i,E}}{\partial e_k} = \left(1 - \frac{1}{N}\right) \cdot \frac{\left(t_{i,k}-\bar{t_i}\right)\sqrt{\sum_{j=1,j\neq k}^{N}\left(e_j-\bar{e}\right)^2 \sum_{j=1}^{N}\left(t_{i,j}-\bar{t_i}\right)^2} - \frac{(e_k-\bar{e})\sum_{j=1}^{N}\left(\left(x_j-\bar{t_i}\right)\left(e_j-\bar{e}\right)\right)\sqrt{\sum_{j=1}^{N}\left(t_{i,j}-\bar{t_i}\right)^2}}{\sqrt{\sum_{j=1}^{N}\left(e_j-\bar{e}\right)^2}}}{\sum_{j=1}^{N}\left(t_{i,j}-\bar{t_i}\right)^2 \sum_{j=1}^{N}\left(e_j-\bar{e}\right)^2} \tag{11}$$

where $U_E$ and $U_{T_i}$ are the measurement uncertainties of thermal error and temperature, respectively. For thermal error, $U_E$ corresponds to the sensor measurement error, which is known. For temperature, however, in addition to sensor error, there is the influence of disturbance information. Disturbance information involves too many factors, such as changes in ambient temperature and heat generated by people. The uncertainty caused by disturbance information is difficult to obtain, but disturbance information and sensor error have a common characteristic: they are independent of thermal error. Therefore, to obtain $U_{T_i}$, a multiple regression model is established, in which the thermal error is the input and the temperature measurement point $T_i$ is the output. Then, the root mean square error (RMSE) of the model is calculated. It represents the standard deviation of the residual data from the temperature measurement data after removing information related to thermal errors. Afterward, it is multiplied by three to get the expanded uncertainty, $U_{T_i}$. The specific calculation method is expressed by the following equation.

$$U_{T_i} = 3\sqrt{\frac{(t_{i,k} - \hat{t}_{i,k})^2}{N-1}} \tag{12}$$

where $\hat{t}_{i,k}$ is the temperature estimate of the model with the thermal error regarded as the input.

Finally, the uncertainty-correlation coefficient is as follows.

$$\rho_{T_i,E}{}^U = \frac{\rho_{T_i,E}}{1 + U_{\rho_{T_i,E}}} \tag{13}$$

Through the ranking of uncertainty-correlation coefficients, the temperature measurement points with the highest ranking are selected as TSPs. The stability TSP selection method shown in Figure 1.

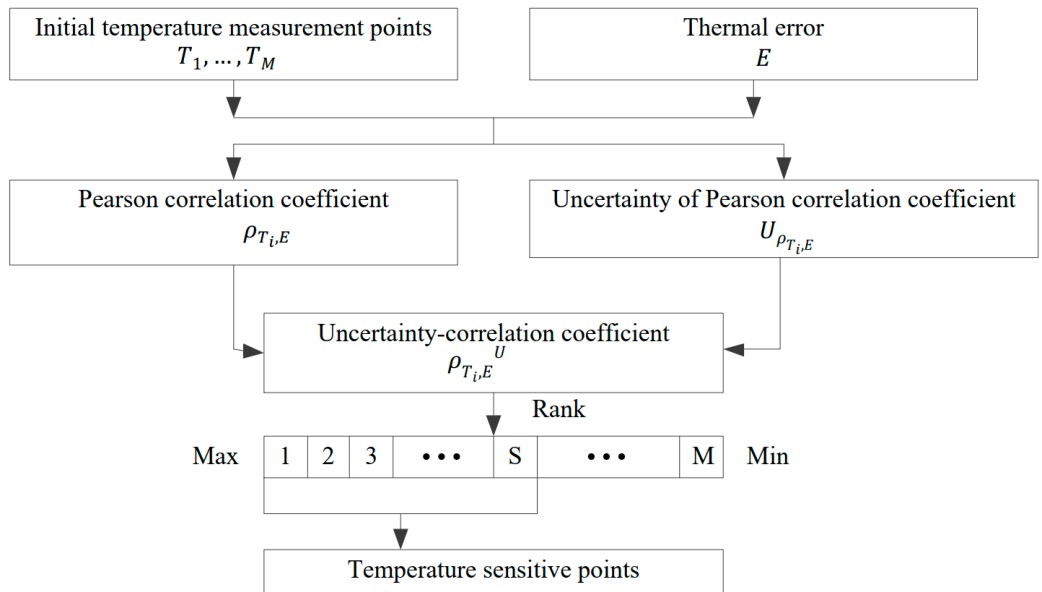

**Figure 1.** Stability temperature-sensitive point (TSP) selection method.

### 3.2. Update Method of Thermal Error Model Based on Update Regression

The multiple linear regression algorithm has a simple form, and the loss function is a convex function with good interpretability and high stability. This study proposed the update regression, an adaptive update strategy for thermal error models based on regression algorithms, that can effectively combine new data with old models.

The algorithm principle is expressed as follows:

Loss function:

$$min : \Gamma = \left( \eta \sum_{i=1}^{N} (\hat{e}_i - e_i)^2 + (1 - \eta) \sum_{j=0}^{S} \left( \beta_j - \beta_j{}^{old} \right)^2 \right) \tag{14}$$

where $\boldsymbol{\beta}^{old} = \left( \beta_0{}^{old}, \dots, \beta_S{}^{old} \right)^{\mathrm{T}}$ is the old model coefficient, $\boldsymbol{\beta} = (\beta_0, \dots, \beta_S)^{\mathrm{T}}$ is the new model coefficient to be solved, and $\eta = [0,1]$ is the weight of the old model. The solution method is as follows.

$$\begin{cases} \frac{\partial \Gamma}{\partial \beta_0} = 0 \\ \quad \vdots \\ \frac{\partial \Gamma}{\partial \beta_S} = 0 \end{cases} \Rightarrow \boldsymbol{\beta} = \left( \eta \boldsymbol{A}^T \boldsymbol{A} + (1 - \eta) \boldsymbol{I} \right)^{-1} \left( \eta \boldsymbol{A}^T \boldsymbol{E} + (1 - \eta) \boldsymbol{\beta}^{old} \right) \tag{15}$$

According to Equations (14) and (15), update regression is a general form of ridge regression. If $\beta^{old} = 0$, then update regression becomes ridge regression. This study found that only a small amount of data and a small $\eta$ are adequate so that the model can be quickly updated and the accuracy of the model can be significantly improved. In this study, $\eta = 0.1$. To use as litlle data as possible to complete the model update, when the difference of the model coefficients established with $N$, $N + 1$, $N + 2$ new data are less than 10%, it is considered that a stable update model has been found. The update method is shown in Figure 2.

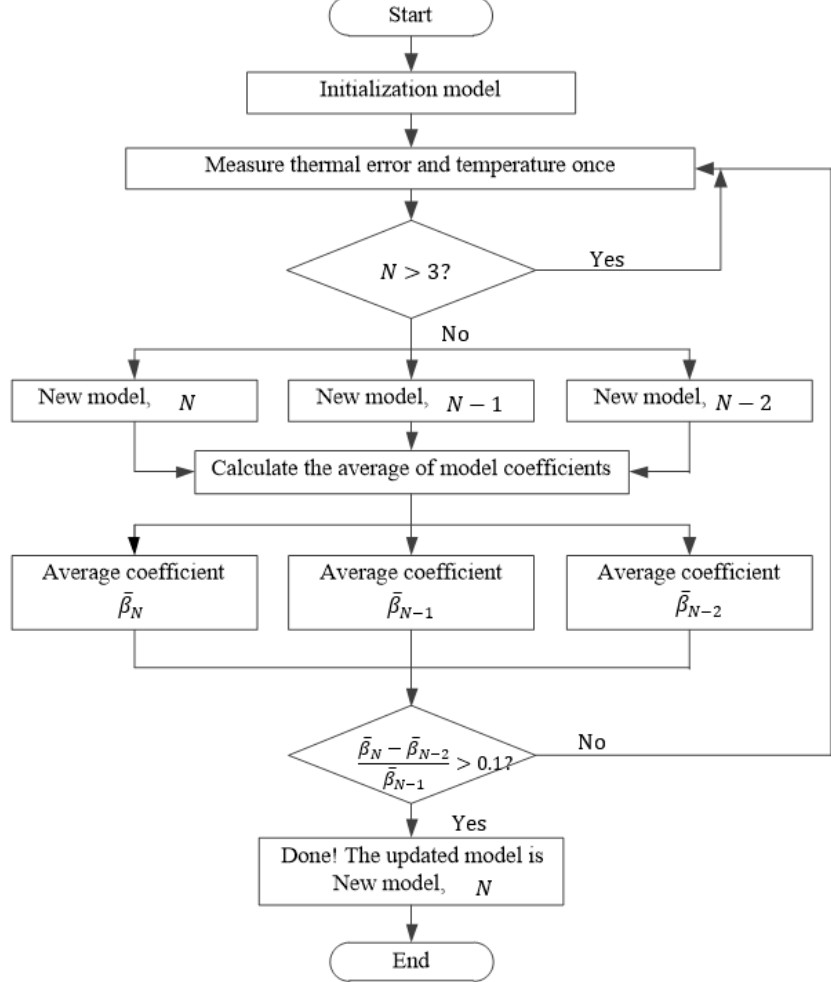

**Figure 2.** Thermal error model update method based on update regression algorithm.

## 4. Experimental Verification

The effect of the proposed method was verified by a long-term experiment, where a specific machine tool was used Leaderway V-450. The Z-axis thermal error was measured in different machining states because the thermal error is largest on the Z-axis.

### 4.1. Experimental Plan

The thermal error is measured by an online detection system, as shown in Figure 3. A cuboid target was installed in the corner of the workbench. To measure the thermal error, the tool was first replaced with the probe and then the probe was allowed to touch the target. In addition, 10 locations at the machine tool, donated by $T_1$, $T_2$, etc., were selected as initial temperature measurement points. $T_1$–$T_9$ were placed near the spindle, and $T_{10}$ was placed on the shell (measuring ambient temperature changes). The details of the temperature sensor are as follows:

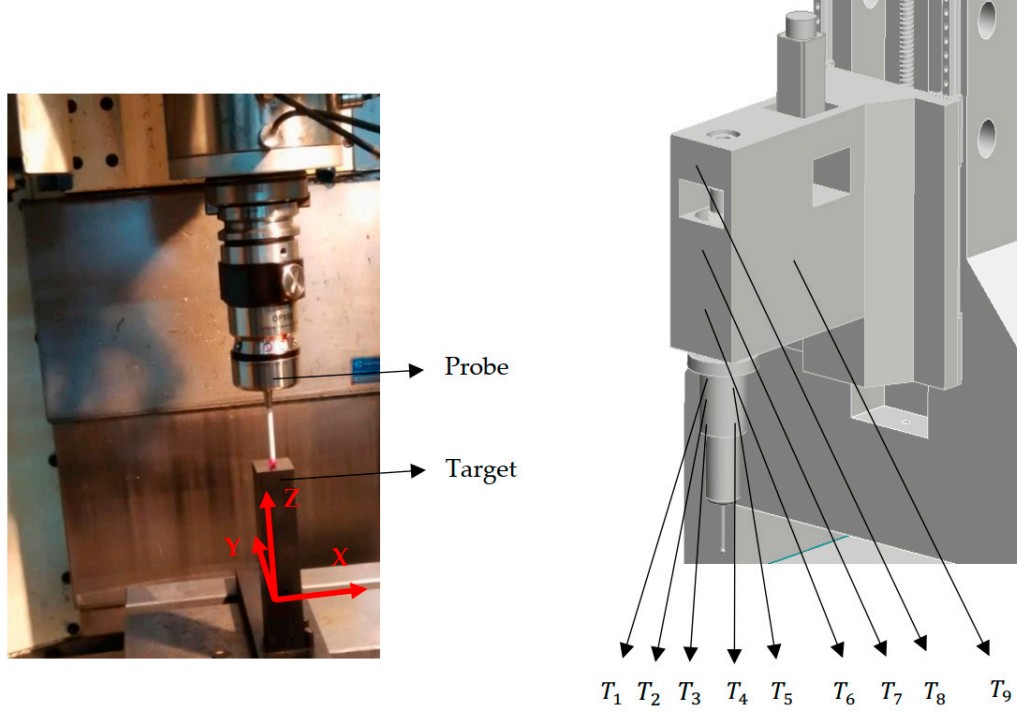

**Figure 3.** Principle of thermal error measurement of machine tool and the positions of temperature measurement points: (**left**) Principle of thermal error measurement; (**right**) Positions of temperature measurement points.

Model: DS18B20, resolution: 0.125 °C, temperature measurement range: −10–80 °C. The temperature sensor is installed by magnetic adsorption: the thermal grease and the sensor are encapsulated in a cylindrical magnet and installed by magnetic force, as shown in Figure 4.

The specific locations are shown in Figure 3.

As shown in Figure 5, the measurement process can be divided into two steps:

Step 1: The probe touches the target, records the coordinates in the Z-axis, and records the values of 10 temperature sensors.

Step 2: The probe is replaced with a cutting tool by the automatic tool change program, the cutting tool starts cutting. Cutting time does not exceed 4 min. Then stops cutting and returns to Step 1.

The material of the workpiece is 45# steel. The material of the cutting tool is ordinary high-speed steel. Steps 1 and 2 take about 4 min. These two steps are repeated for 4 h. In addition, when cutting, the tool needs to move back and forth, and when changing direction, it also stops cutting for a short time. Therefore, the wear of the tool is not serious. In addition, tool wear does not affect the measurement accuracy of thermal errors, because thermal errors require the tool to be replaced with a probe for measurement. The first coordinate and temperature measurements were regarded as the initial values. To obtain the thermal error and temperature change, the initial value was subtracted from each measurement. Twenty batches of experiments, denoted by B1, B2, etc., were performed. The experimental parameters are listed in Table 1.

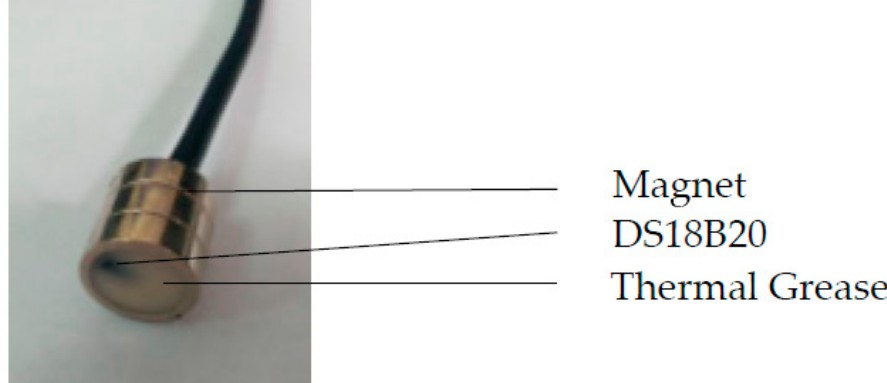

**Figure 4.** Temperature sensor.

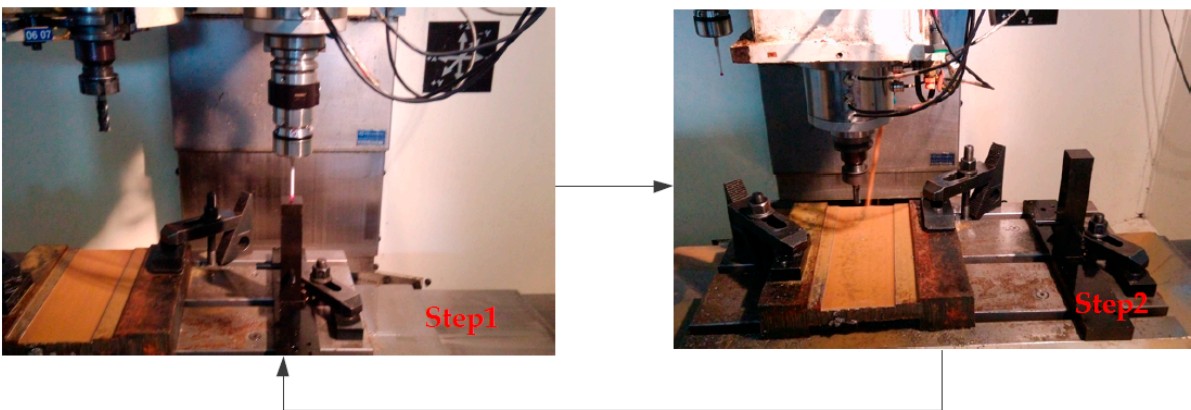

**Figure 5.** Thermal error measurement process.

**Table 1.** Experimental parameters.

| Batch | Rotating Speed (rpm) | Feed (mm/min) | Cutting Depth (μm) |
|-------|----------------------|---------------|--------------------|
| B1 | 1000 | 400 | 100 |
| B2 | 500 | 600 | 100 |
| B3 | 1500 | 600 | 50 |
| B4 | 1500 | 600 | 100 |
| B5 | 1500 | 600 | 100 |
| B6 | 1000 | 600 | 50 |
| B7 | 1000 | 600 | 150 |
| B8 | 1500 | 400 | 150 |
| B9 | 500 | 400 | 50 |
| B10 | 1000 | 600 | 100 |
| B11 | 500 | 400 | 50 |
| B12 | 1000 | 600 | 150 |
| B13 | 1500 | 800 | 100 |
| B14 | 1000 | 600 | 150 |
| B15 | 1500 | 800 | 100 |
| B16 | 1000 | 400 | 100 |
| B17 | 800 | 500 | 150 |
| B18 | 1500 | 600 | 50 |
| B19 | 1000 | 400 | 100 |
| B20 | 1500 | 800 | 100 |

### 4.2. Experimental Result

To save space, only the experimental data of B1, B10, and B20 are shown in Figure 6.

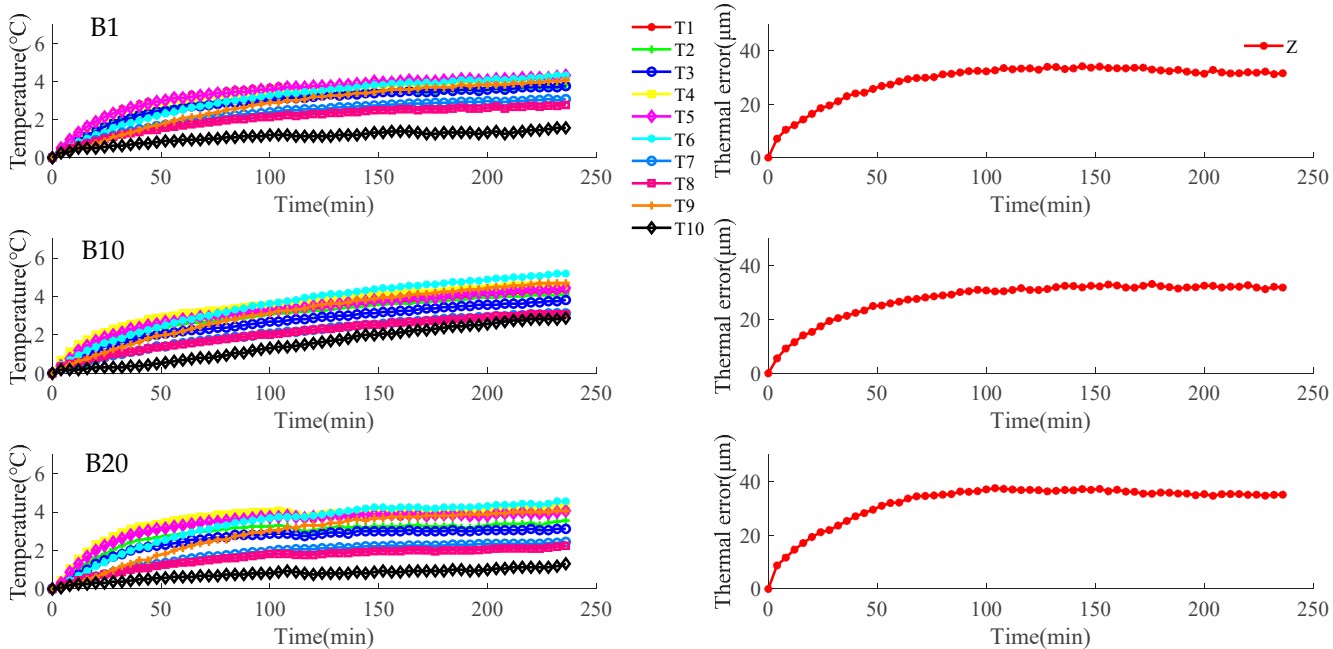

**Figure 6.** Measurement data of B1, B10, and B20.

In Figure 6, the horizontal axis represents time, and the vertical axis represents the variation of temperature (left) and thermal error (right) of the 10 sensors, respectively. Variation means the increment of each measurement relative to the initial measurement. It can be seen that with the change of time, the temperature and thermal errors change obviously at first, and then gradually become stable and reach thermal equilibrium.

#### 4.2.1. Comparison Results of TSP Selection Methods

Each batch of experimental data was used for the selection of the corresponding TSPs. Furthermore, according to previous test results on the machine tools, there is no obvious difference in the effect of two and more TSPs [22]. Thus, for the convenience of analysis, this study reduced the number of TSPs to two. There are two TSP selection methods for comparison.

(1)  Fuzzy clustering and correlation coefficient. First, the initial temperature measurement points are divided into two groups. Then, for each group, the point with the highest correlation is selected as a TSP.

(2)  Stability TSP selection method. The two points with the highest uncertainty-correlation coefficients are selected as TSPs.

The TSP selection results are presented in Table 2.

According to Table 2, for the fuzzy clustering and correlation coefficient, the most frequent TSP of the first group is mainly $T_1$ or $T_5$, followed by $T_7$ or $T_4$; for the other group, the most frequent TSP is $T_{10}$, followed by $T_6$, $T_7$ or $T_8$. For the stability TSP selection method, the most frequent TSP combinations are $T_1$ and $T_5$, followed by $T_1$ and $T_4$. The TSPs selected by the stability TSP selection method have less variability.

**Table 2.** TSP selection results.

| Batch | TSPs Selected by Fuzzy Clustering and Correlation Coefficient | TSPs Selected by Stability TSP Selection Method |
|---|---|---|
| B1 | $T_1, T_{10}$ | $T_1, T_5$ |
| B2 | $T_5, T_{10}$ | $T_1, T_5$ |
| B3 | $T_7, T_4$ | $T_1, T_5$ |
| B4 | $T_1, T_7$ | $T_1, T_5$ |
| B5 | $T_4, T_{10}$ | $T_1, T_4$ |
| B6 | $T_1, T_{10}$ | $T_1, T_5$ |
| B7 | $T_4, T_{10}$ | $T_1, T_4$ |
| B8 | $T_7, T_{10}$ | $T_1, T_5$ |
| B9 | $T_7, T_{10}$ | $T_1, T_5$ |
| B10 | $T_4, T_{10}$ | $T_1, T_4$ |
| B11 | $T_5, T_{10}$ | $T_1, T_5$ |
| B12 | $T_1, T_{10}$ | $T_1, T_5$ |
| B13 | $T_5, T_{10}$ | $T_1, T_5$ |
| B14 | $T_1, T_{10}$ | $T_1, T_5$ |
| B15 | $T_5, T_{10}$ | $T_1, T_5$ |
| B16 | $T_1, T_8$ | $T_1, T_5$ |
| B17 | $T_1, T_4$ | $T_1, T_5$ |
| B18 | $T_5, T_8$ | $T_1, T_5$ |
| B19 | $T_1, T_6$ | $T_1, T_5$ |
| B20 | $T_5, T_{10}$ | $T_1, T_5$ |

4.2.2. Comparison Results of Modeling Accuracy of TSP Selection Methods

Afterward, the modeling accuracy of the TSPs selected by the two methods were compared. The thermal error model was built for each batch of data of B1–B20. The models established by B1–B20 data are recorded as M1–M20. The prediction accuracy of M1–M20 was calculated. Six modeling methods were involved in the comparison. That is, the model Mi established by different methods is also different.

(1) Clustering + Correlation coefficient + Rig: Fuzzy clustering and correlation coefficient for TSP selection, and ridge regression for modeling.
(2) Stability TSP selection method + Rig: Stability TSP selection method for TSP selection, and ridge regression for modeling.
(3) Clustering + Correlation coefficient + Reg: Fuzzy clustering and correlation coefficient for TSP selection, and ordinary multiple regression for modeling.
(4) Stability TSP selection method + Reg: Stability TSP selection method for TSP selection, and the ordinary multiple regression for modeling.
(5) Clustering + Correlation coefficient + NN: Fuzzy clustering and correlation coefficient for TSP selection, and neural network for modeling.
(6) Stability TSP selection method + NN: Stability TSP selection method for TSP selection, and neural network for modeling.

The NN structure was 2-4-3-1, one input layer with two input nodes, two hidden layers with four and three nodes, respectively, and one output layer with one output node. The activation function of the hidden layer was sigmoid, and the activation function of input and output layers was purely linear.

Based on the RMSE, three levels of prediction accuracy indicators were calculated, and the calculation method is as shown in Equation (16). For ease of understanding, the relationship between the three levels of accuracy indicators is shown in Figure 7.

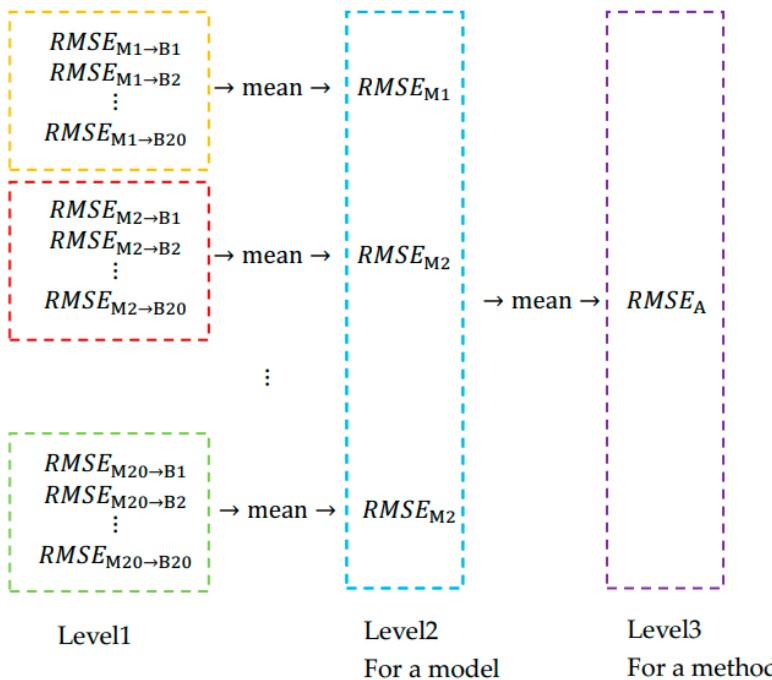

**Figure 7.** Relationship of three levels of prediction accuracy indicators.

Level 1: $RMSE_{Mi \to Bj}$ is the RMSE of M$i$ predicting data B$j$.

Level 2: $RMSE_{Mi}$ is the mean value of the $RMSE_{Mi \to Bj}$ for each model, reflecting the accuracy of a model.

Level 3: $RMSE_A$ is the mean value of the $RMSE_{Mi}$ for each modeling algorithm, reflecting the accuracy of a modeling methods.

$$RMSE_{Mi \to Bj} = \sqrt{\frac{\sum_{k=1}^{N}(e_k - \hat{e}_k)^2}{N}} \quad RMSE_{Mi} = \frac{\sum_{j=1}^{20} RMSE_{Mi \to Bj}}{20} \quad RMSE_A = \frac{\sum_{i=1}^{20} RMSE_{Mi}}{20} \tag{16}$$

where $e_k$ is the $k$-th measurement in the $B_j$ and $\hat{e}_k$ is the corresponding predicted value.

The comparison results are shown in Figure 8 and Table 3.

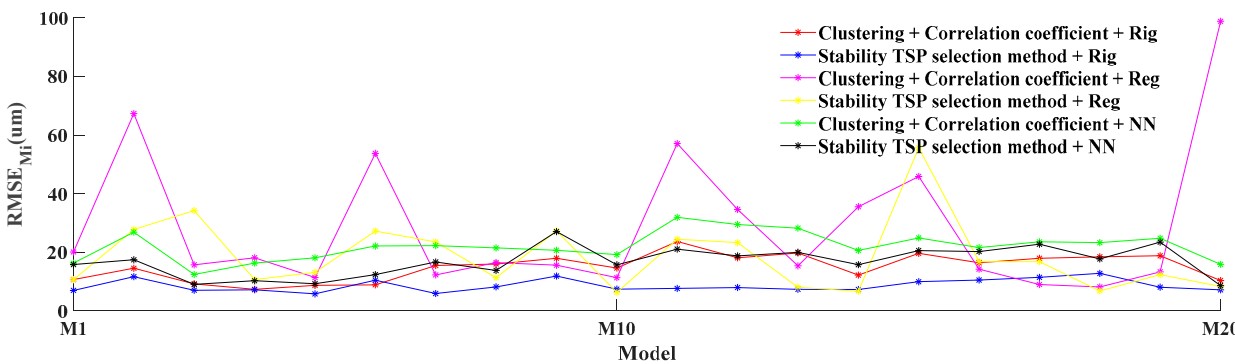

**Figure 8.** $RMSE_{Mi}$ of models established by different methods (for TSP selection effect comparison).

Among the three modeling methods, the accuracy of the TSPs selected by the stability TSP selection method is better than that of the fuzzy clustering and correlation coefficient. The accuracy of the ridge regression is better than the NN and the ordinary multiple regression.

**Table 3.** $RMSE_A$ of different modeling methods (for TSP selection effect comparison).

| Modeling Methods | $RMSE_A$ µm |
|---|---|
| Clustering + Correlation coefficient + Rig | 14.95 |
| Stability TSP selection method + Rig | 8.65 |
| Clustering + Correlation coefficient + Reg | 28.7 |
| Stability TSP selection method + Reg | 18.6 |
| Clustering + Correlation coefficient + NN | 21.9 |
| Stability TSP selection method + NN | 16.4 |

4.2.3. Comparison Results of the Adaptive Model Update Algorithm

To verify the effect of the proposed model update method, the following three methods were compared by $RMSE_{Mi}$. The three methods differ in whether and how to update the model with new data.

(1)  No update model ("No update" in Figure 8): Models M1–M20 are established by B1–B20, respectively. The modeling method is "Stability TSP selection method + Rig". No updates are made when predicting the new data.

(2)  Update model by new data + old model ("New data + old model" in Figure 8): When predicting the data B$j$, the old model and new data are both used to build the model. The old model refers to M1–M20 of the "No update" model. A segment of data B$j$ from beginning is used for the updating model as new data. The model updating algorithm is update regression. The Mi of $RMSE_{Mi}$ means old model.

(3)  Update model only by new data ("New data-$N$" in Figure 8, where $N$ is the length of the new data): When predicting the data B$j$, a segment of data B$j$ from the beginning is used to build the model as new data. The lengths of the new data are 3, 5, 7, 9, 11, ..., 25, 60, respectively. The modeling method is "Stability TSP selection method + Rig". Since no old model is involved in prediction, $RMSE_{M1} = RMSE_{M2} = \cdots = RMSE_{M20}$.

The result is shown in Figure 9.

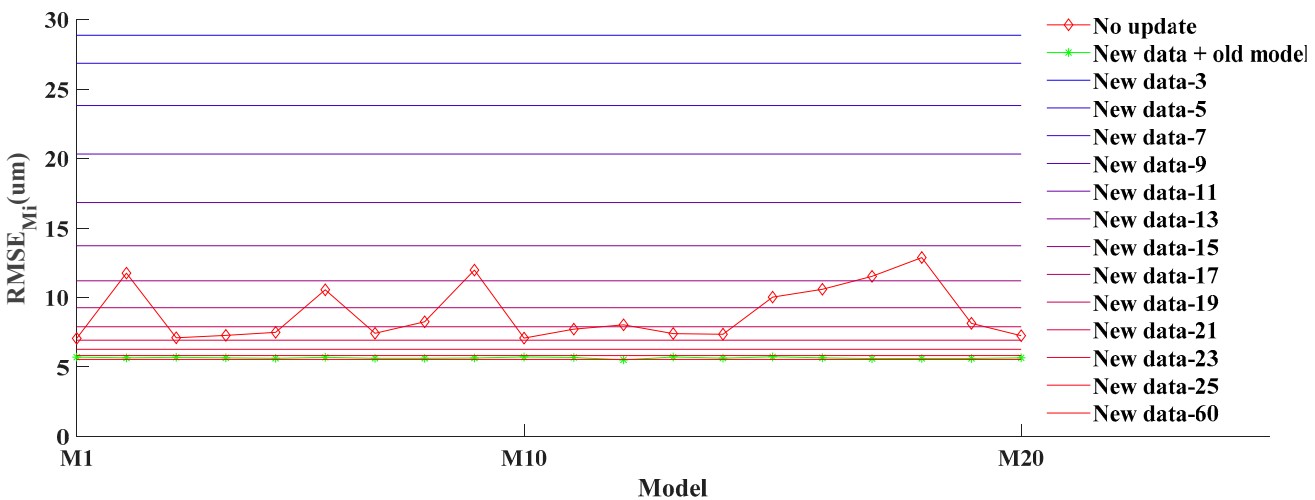

**Figure 9.** $RMSE_{M_i}$ of models established by different methods (for model update method comparison).

The "Update model by new data + old model" requires an average of six new data for model update. In Figure 9, the accuracy of the "Update model by new data + old model" is higher than that of the "No update" model and close to that of the "Update model only by new data" with 60 new data lengths. The 60 data length means that all the predicted data are used for modeling. The $RMSE_{Mi}$ does not indicate the prediction accuracy but the fitting accuracy. This means that the model can almost have the highest accuracy a regression model can achieve after updating.

### 4.3. Validation of the Proposed Model

According to the measurement data of B1–B20, the update effect of the model in application is validated. In addition, the computational difficulty of the model update is tested.

### 4.3.1. Model Precision

The goal is to keep thermal errors within 10 μm. The model update needs an old model and the old model was built by data of B1. The data of B2~B20 are concatenated as a data of 76 h and used for predicting. Then start predicting from time 0. When it is detected that the residual error of the model prediction is greater than 7 μm, the model is updated and the prediction is continued to the end. The result is shown in Figure 10.

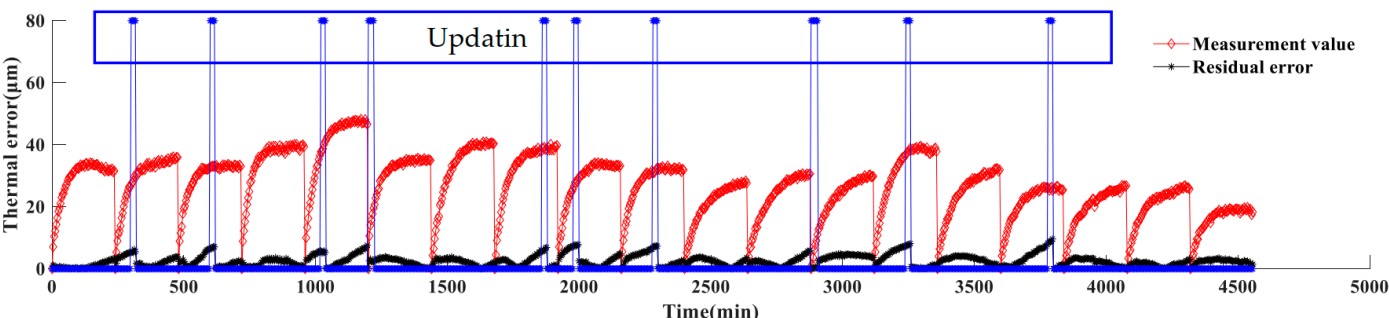

**Figure 10.** Model precision validation.

The thermal error measurements and residual error after model prediction and compensation are shown in Figure 10. It can be seen that the residual error is significantly reduced by model updating when it is too large, and the prediction error of the model in the whole process does not exceed 10 μm. During the whole period, it lasted for a total of 4560 min, and 10 updates were carried out, which consumed a total of 184 min, accounting for 4% of the total time.

This model verification is reliable because the thermal error compensation technology is mature. Compensation can be performed quickly through the built-in origin offset function of the CNC system. Compensation accuracy depends only on the accuracy of the thermal error model.

### 4.3.2. Computation Time of Model Update

In order to reduce the cost as much as possible, the embedded platform is chosen to test the calculation time of the model update. In the model update process, the most complex operation is the update regression algorithm, because it involves the operation of the matrix. According to previous experimental results, each update requires an average of six batches of data. Based on this, the entire process of the model update is rewritten in C language. The TMS320F28335 DSP processor is used as an embedded platform to run the program, and the CPU clock frequency is 150 Mhz. During the calculation process, the timer is used for timing, and the timing resolution is 0.1 ms. The results show that the time for each model update is about 130 μs running in FLASH, which can be negligible. The computation time can be further shortened if the program is moved to RAM. Therefore, the computational difficulty of updating the model is very low, and it can be completed by ordinary embedded platforms.

Relevant project files will be uploaded together with the paper as Video S1 (Supplementary Materials): Running time test on DSP processor.

## 5. Discussion

In this section, the above experimental results are discussed and analyzed.

### 5.1. Discussion of Comparison Results of TSP Selection Methods

The TSPs selected by the "Stability TSP selection method" were more stable than those selected by the "Fuzzy clustering and correlation coefficient." To analyze the reasons, the three kinds of data were calculated.

(1)  For each temperature measurement point, the mean correlation of all batch data was calculated by two correlation calculation methods shown in the following equation. This process indicates the long-term correlation. The results are shown in Figure 11.

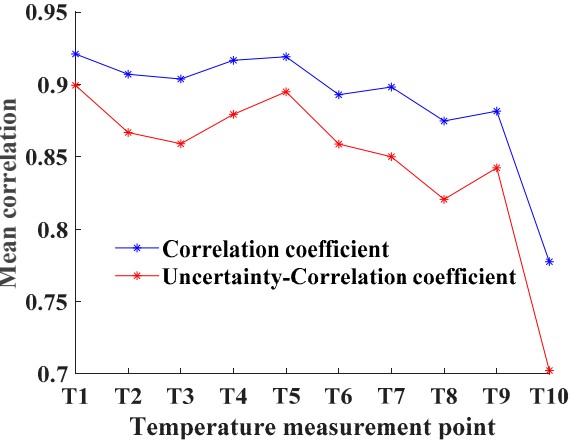

**Figure 11.** Mean correlation of temperature measurement point.

Mean uncertainty-correlation coefficient:

$$\frac{\sum_{j=1}^{20} \left( \rho_{T_i, E}{}^U{}_{Bj} \right)}{20} \tag{17}$$

Mean correlation coefficient:

$$\frac{\sum_{j=1}^{20} \left( \rho_{T_i, EBj} \right)}{20} \tag{18}$$

(2)  For each temperature measurement point, the standard deviation (*Std*) and mean uncertainty of the correlation coefficient of all batch data were calculated. The specific calculation methods are expressed by the following equation. The results are depicted in Figure 12. This process indicates the stability of the correlation.

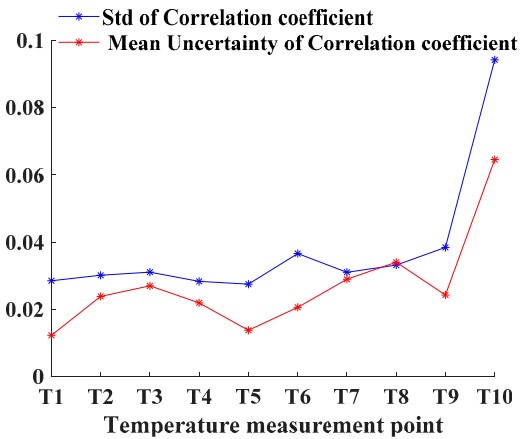

**Figure 12.** Standard deviation (Std) and mean uncertainty of the correlation coefficient of temperature measurement point.

Std of the correlation coefficient:

$$Std_{j=1}^{20}\left(\rho_{T_i,EBj}\right) \tag{19}$$

Mean uncertainty of the correlation coefficient:

$$\frac{\sum_{j=1}^{20}\left(U_{T_iBj}\right)}{20} \tag{20}$$

(3) The fuzzy clustering results of T1–T10 in B1–B20, are listed in Table 4.

**Table 4.** Grouping results of temperature measurement points of B1–B20.

| Batch | Group1 | Group2 | Batch | Group1 | Group2 |
|-------|--------|--------|-------|--------|--------|
| B1 | $T_1$–$T_9$ | $T_{10}$ | B2 | $T_1$–$T_9$ | $T_{10}$ |
| B3 | $T_1$, $T_5$, $T_7$, $T_8$, $T_{10}$ | $T_2$–$T_4$, $T_6$, $T_9$ | B4 | $T_1$–$T_5$ | $T_6$–$T_{10}$ |
| B5 | $T_1$–$T_9$ | $T_{10}$ | B6 | $T_1$–$T_9$ | $T_{10}$ |
| B7 | $T_1$–$T_9$ | $T_{10}$ | B8 | $T_1$–$T_9$ | $T_{10}$ |
| B9 | $T_1$–$T_9$ | $T_{10}$ | B10 | $T_1$–$T_9$ | $T_{10}$ |
| B11 | $T_1$–$T_9$ | $T_{10}$ | B12 | $T_1$–$T_9$ | $T_{10}$ |
| B13 | $T_1$–$T_9$ | $T_{10}$ | B14 | $T_1$–$T_9$ | $T_{10}$ |
| B15 | $T_1$–$T_9$ | $T_{10}$ | B16 | $T_1$–$T_7$, $T_9$ | $T_8$, $T_{10}$ |
| B17 | $T_1$–$T_7$, $T_9$ | $T_8$, $T_{10}$ | B18 | $T_1$, $T_5$, $T_7$, $T_8$, $T_{10}$ | $T_2$–$T_4$, $T_6$, $T_9$ |
| B19 | $T_1$–$T_5$, $T_7$–$T_{10}$ | $T_6$ | B20 | $T_1$–$T_9$ | $T_{10}$ |

The following conclusions can be drawn from the above calculation data.

(1) The low collinearity and high correlation of TSPs are conflicting, resulting in a low correlation for one of the TSPs selected by the "Fuzzy clustering and correlation coefficient". There is no grouping process in the "Stability TSP selection method", so high correlation TSPs can be selected. According to Figure 11, it can be seen that the correlation of $T_1$–$T_9$ is significantly higher, and the correlation of $T_{10}$ is lower. According to Table 4, in most cases, the "Fuzzy clustering and correlation coefficient" method will group $T_1$–$T_9$ into the same group, indicating that the collinearity between $T_1$ and $T_9$ is also high. This confirms the theoretical analysis in the introduction that the correlation of temperature measurement points is transitive. Therefore, reducing the collinearity of TSPs and improving the correlation at the same time are in conflict. This results in the "Fuzzy clustering and correlation coefficient" method always grouping $T_{10}$ alone, eventually causing one of the TSPs to have a low correlation. According to Figure 12, the correlation of temperature measurement points with low correlation is also more unstable. Since correlation is the basis for the selection of TSPs, low correlation is one of the reasons for the variability of TSPs. The "Stability TSP selection method" directly selects TSPs with a strong correlation, so this problem does not exist.

(2) The uncertainty of the correlation coefficient is an effective indicator to evaluate the stability of the correlation of the temperature measuring points, and it can be used to select TSPs with more stable correlation. According to Figure 12, the Std and uncertainty of the correlation coefficient have the same effect on measuring stability. However, there is an important difference between them. In particular, the Std calculation of the correlation coefficient needs all batches of data, which means multiple long-term experiments with the machine. The uncertainty of the correlation coefficient can be calculated only by one batch of data. This means that the uncertainty of the correlation coefficient can calculate the stability of the correlation in a short time. Furthermore, according to Figure 11, when the correlation of temperature measurement points is similar, the uncertainty-correlation coefficient promotes the selection of TSPs with a stable correlation, thereby significantly reducing the variability of TSPs.

## 5.2. Discussion of Comparison Results of Modeling Accuracy of TSP Selection Methods

The TSPs selected by the "Stability TSP selection method" have high correlation and stability, but also high collinearity. For unbiased estimation algorithms, such as ordinary multiple regression, high collinearity will amplify the variance of model coefficient estimates, resulting in model instability. However, with biased regression, such as ridge regression, the collinearity problem can be significantly suppressed [38]. According to Figure 8 and Table 3, the long-term prediction accuracy of the thermal error model built by stable TSPs is higher. The reason can be revealed by the model coefficients. Figure 13 exemplifies the coefficients of the models established with stable TSPs ($T_1$, $T_5$) and unstable TSPs ($T_6$, $T_{10}$) based on the experimental data of B1–B20. The modeling algorithm is ridge regression. The model coefficients of stable TSPs vary slightly around the value 5. The model coefficients of unstable TSPs vary between the values 0 and 10, with significantly larger changes. This means that the connection of unstable TSPs with thermal error is more unstable.

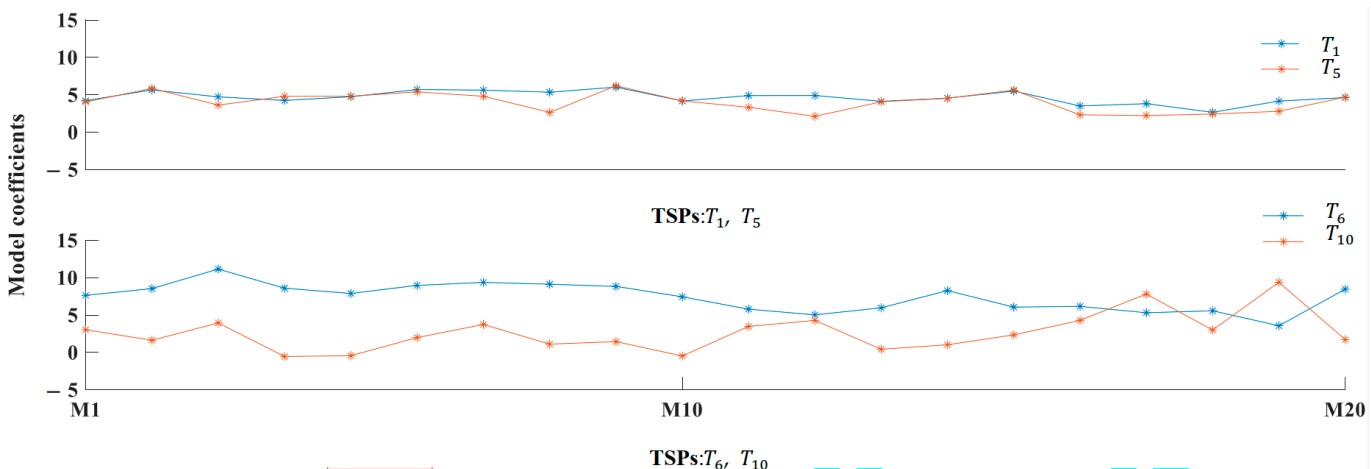

**Figure 13.** Model coefficients of stable TSPs ($T_1$, $T_5$) and unstable TSPs ($T_6$, $T_{10}$).

## 5.3. Discussion of Comparison Results of the Adaptive Model Update Algorithm

In this section, the reason why the update regression can complete the model update with six data is analyzed. To illustrate this, the following data need be calculated.

Models M1–M20 (TSPs: $T_1$, $T_5$, ridge regression, 60 data) were used to predict the first six data of B1–B20, and the RMSE, donated by $RMSE6_{Mi \to Bj}$ was calculated. The thermal error model built by 60 data represents the thermal error law in one experiment, and $RMSE6_{Mi \to Bj}$ represents the fit degree between the first six data and the thermal error model. The results are presented in Figure 14.

In Figure 14, the diagonal line is $RMSE6_{Mi \to Bj}(i = j)$, which was calculated by the model predicting the first six modeling data. For ease of description, it is referred to as "self-fit degree". Obviously, for the first six data, the self-fit degree is generally higher than the fit degree of other models. This shows that the thermal error information contained in the first six data is closer to the thermal error data of the same batch. Therefore, in theory, as long as an appropriate mathematical algorithm is selected, the thermal error law can be extracted from the first six data to update the model and improve the prediction accuracy. However, the sample size is very small, which is not conducive to modeling. If a modeling algorithm with high fitting accuracy is used, such as multiple regression, neural network, errors in the data may cause severe overfitting. The partial regression algorithm can solve the overfitting problem by adding a regularization term to the loss function. The typical algorithm is ridge regression (as shown in Equation (14), where $\beta^{old} = 0$). However, the regularization term of ridge regression makes the model coefficients tend to zero. For small sample data, this trend is more severe, because the proportion of the regularization term in the loss function will relatively increase and cause underfitting. According to Figure 13,

with a change in the machine processing parameters, the thermal error model coefficient changes but not significantly. Therefore, even if the old model fails, its model coefficients do not deviate too much from the accuracy coefficients. Hence, in the loss function, it is obviously a better choice to make the model coefficients tend to the old model coefficients. This can solve the overfitting problem through the regularization term. Simultaneously, it greatly reduces the risk of underfitting. As shown in Figure 15, the effect of update regression can be intuitively understood by the model coefficients. Figure 15 includes the coefficients of two kinds of models: (1) the model built by ridge regression with 60 data and (2) the model built by the update regression with six data. The model coefficients of the two algorithms are very close. This shows that update regression can extract accurate thermal error laws from six data.

|  | B1 | B2 | B3 | B4 | B5 | B6 | B7 | B8 | B9 | B10 | B11 | B12 | B13 | B14 | B15 | B16 | B17 | B18 | B19 | B20 |
|---|---|---|---|---|---|---|---|---|---|---|---|---|---|---|---|---|---|---|---|---|
| **M1** | 0.48 | 10.08 | 8.94 | 0.87 | 9.53 | 9.82 | 9.10 | 8.77 | 11.31 | 3.40 | 7.53 | 6.23 | 1.70 | 0.17 | 8.40 | 2.29 | 3.27 | 4.07 | 0.58 | 4.30 |
| **M2** | 13.65 | 3.26 | 4.58 | 16.03 | 6.89 | 3.59 | 1.18 | 3.97 | 6.34 | 11.33 | 3.04 | 3.01 | 9.08 | 13.20 | 3.09 | 9.01 | 7.08 | 12.52 | 8.29 | 12.44 |
| **M3** | 8.08 | 7.46 | 4.52 | 9.05 | 4.06 | 5.74 | 5.78 | 4.41 | 9.27 | 5.81 | 5.48 | 2.82 | 5.11 | 7.75 | 6.32 | 5.08 | 3.53 | 8.54 | 5.24 | 6.11 |
| **M4** | 1.57 | 10.20 | 9.35 | 0.80 | 9.80 | 10.09 | 9.44 | 9.37 | 11.41 | 3.83 | 7.90 | 6.83 | 2.25 | 1.11 | 8.50 | 3.12 | 4.03 | 3.58 | 1.97 | 4.73 |
| **M5** | 8.81 | 7.00 | 3.99 | 10.32 | 2.90 | 5.09 | 5.44 | 4.20 | 8.92 | 6.59 | 5.46 | 2.89 | 5.59 | 8.55 | 5.97 | 5.37 | 3.65 | 9.00 | 5.30 | 7.04 |
| **M6** | 11.04 | 5.61 | 0.57 | 12.59 | 2.88 | 2.32 | 3.33 | 0.23 | 7.90 | 8.77 | 4.20 | 1.03 | 7.26 | 10.61 | 4.87 | 7.31 | 5.64 | 10.65 | 7.03 | 9.52 |
| **M7** | 10.77 | 5.85 | 0.28 | 12.03 | 2.29 | 2.72 | 3.47 | 0.31 | 8.07 | 8.48 | 4.14 | 1.29 | 7.09 | 10.29 | 5.05 | 7.26 | 5.68 | 10.49 | 7.09 | 9.18 |
| **M8** | 6.46 | 8.49 | 5.05 | 4.75 | 6.15 | 6.91 | 6.04 | 3.63 | 10.06 | 3.87 | 4.86 | 0.78 | 4.20 | 5.69 | 7.10 | 5.19 | 4.32 | 7.73 | 6.05 | 3.73 |
| **M9** | 16.48 | 2.04 | 7.75 | 19.43 | 10.40 | 7.13 | 4.83 | 7.04 | 4.20 | 14.02 | 0.52 | 5.15 | 11.08 | 15.94 | 0.96 | 11.04 | 8.91 | 14.64 | 9.97 | 15.50 |
| **M10** | 3.46 | 9.36 | 7.84 | 3.87 | 8.16 | 8.76 | 8.24 | 7.71 | 10.74 | 0.55 | 7.01 | 5.43 | 1.46 | 3.40 | 7.83 | 0.81 | 1.85 | 5.53 | 2.07 | 1.58 |
| **M11** | 11.27 | 5.59 | 2.01 | 12.06 | 3.12 | 1.81 | 2.51 | 2.70 | 7.88 | 8.93 | 3.34 | 2.92 | 7.53 | 10.65 | 4.84 | 8.00 | 6.56 | 10.95 | 7.92 | 9.66 |
| **M12** | 4.15 | 9.32 | 6.86 | 2.03 | 7.92 | 8.35 | 7.42 | 5.91 | 10.71 | 0.89 | 5.93 | 3.36 | 2.40 | 3.53 | 7.77 | 3.18 | 2.22 | 6.21 | 4.37 | 0.87 |
| **M13** | 2.03 | 9.75 | 8.42 | 2.14 | 8.90 | 9.32 | 8.69 | 8.26 | 11.05 | 2.31 | 7.28 | 5.84 | 0.48 | 2.05 | 8.13 | 1.29 | 2.63 | 4.79 | 1.16 | 3.18 |
| **M14** | 0.84 | 9.97 | 8.77 | 0.71 | 9.32 | 9.66 | 8.97 | 8.61 | 11.22 | 3.05 | 7.46 | 6.11 | 1.36 | 0.97 | 8.31 | 2.00 | 3.08 | 4.32 | 0.19 | 3.94 |
| **M15** | 13.68 | 3.29 | 4.75 | 15.84 | 6.88 | 3.65 | 1.48 | 4.35 | 6.36 | 11.33 | 2.77 | 3.42 | 9.13 | 13.18 | 3.10 | 9.17 | 7.32 | 12.58 | 8.52 | 12.44 |
| **M16** | 3.35 | 10.73 | 9.66 | 4.76 | 10.69 | 10.67 | 9.64 | 9.23 | 11.84 | 5.07 | 7.70 | 6.39 | 3.05 | 3.32 | 8.91 | 3.23 | 3.76 | 2.70 | 1.41 | 6.11 |
| **M17** | 2.57 | 10.56 | 9.23 | 4.49 | 10.35 | 10.35 | 9.28 | 8.64 | 11.71 | 4.58 | 7.35 | 5.81 | 2.51 | 2.72 | 8.77 | 2.41 | 3.00 | 3.34 | 0.68 | 5.60 |
| **M18** | 6.38 | 11.69 | 11.28 | 7.55 | 12.43 | 12.12 | 10.96 | 11.00 | 12.61 | 7.34 | 8.67 | 7.86 | 5.06 | 6.08 | 9.68 | 5.53 | 5.71 | 2.17 | 4.06 | 8.57 |
| **M19** | 3.32 | 10.73 | 9.63 | 4.82 | 10.68 | 10.65 | 9.61 | 9.16 | 11.84 | 5.06 | 7.66 | 6.32 | 3.02 | 3.31 | 8.91 | 3.15 | 3.67 | 2.74 | 1.25 | 6.10 |
| **M20** | 3.73 | 9.27 | 7.72 | 4.25 | 8.00 | 8.64 | 8.16 | 7.62 | 10.67 | 0.44 | 6.98 | 5.38 | 1.69 | 3.67 | 7.76 | 1.07 | 1.72 | 5.67 | 2.18 | 1.12 |

**Figure 14.** $RMSE6_{Mi \rightarrow Bj}$ of M1–M20 (TSPs: $T_1$, $T_5$, ridge regression, 60 data) for the prediction of the first six data of B1–B20.

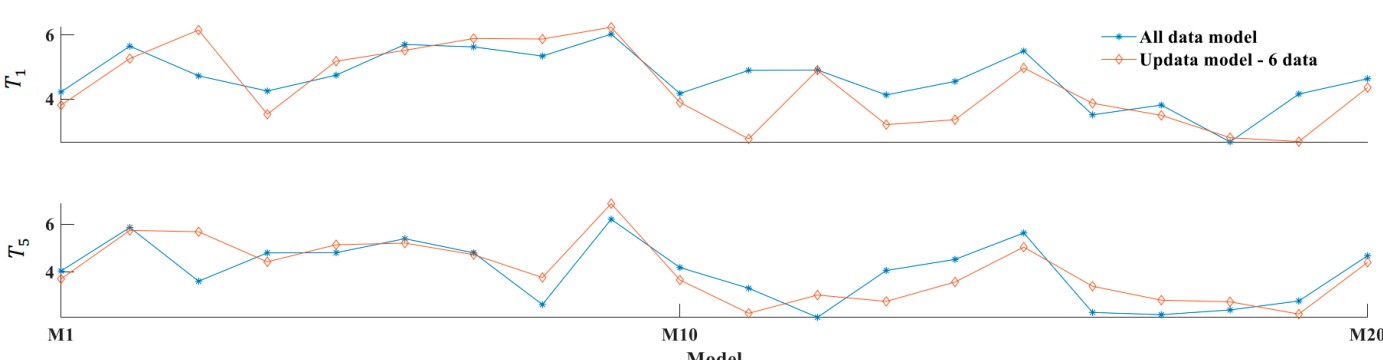

**Figure 15.** Coefficients of two kinds of models: (1) the model built by ridge regression with 60 data and (2) the model built by the update regression with six data.

## 6. Conclusions

This study investigated the selection of TSPs and the adaptive update method of the thermal error model of CNC machine tools. The conclusions are as follows.

(1) When selecting the TSPs, the grouping method could cause the conflict of the collinearity and correlation of the TSPs and result in the low correlation of the selected TSPs. The stability of the correlation should also be considered when performing the correlation calculation in TSP selection. These two problems lead to the variability of TSPs, which reduces the prediction accuracy of thermal error models. This means that the

TSPs selection method is unstable and may lead to inappropriate TSPs. In this regard, a stability TSP selection method was proposed in this study. In the proposed method, the correlation is the primary basis for the selection of TSPs. Furthermore, when calculating the correlation, the stability of correlation is calculated simultaneously so that the TSPs with high and stable correlations can be selected. After experimental verification, the variability of TSPs selected by the proposed method almost disappeared. In addition, for a variety of thermal error modeling algorithms, higher thermal error prediction accuracy could be obtained.

(2) This study proposed a new thermal error model adaptive update method. This approach adds a regularization term that tends to the old model coefficients in the loss function of the regression algorithm. Based on the new loss function, the new data can be fused with the old model. Many experiments verified by that the proposed method could complete the update of the thermal error model with an average of six new data. Moreover, the prediction accuracy was close to the highest accuracy (fitting accuracy) of the linear regression algorithm, which greatly reduced the model update time and improved the accuracy of the updated model.

**Supplementary Materials:** The following supporting information can be downloaded at: https://www.mdpi.com/article/10.3390/machines10060427/s1, Extraction code: j2xy. Video S1: Running time test on DSP processor.

**Author Contributions:** Conceptualization, H.L.; methodology, H.L.; software, H.L.; validation, H.L.; formal analysis, L.Z.; investigation, H.L.; resources, E.M. and J.W.; data curation, E.M.; writing—original draft preparation, H.L.; writing—review and editing, L.Z., S.Z. and J.W.; visualization, H.L.; supervision, E.M.; project administration, E.M.; funding acquisition, H.L. and E.M. All authors have read and agreed to the published version of the manuscript.

**Funding:** This research was funded by the National Natural Science Foundation of China (grant number: 52105274), the National Key R&D Program of China (grant number: 2019YFB1703700), and the Natural Science Foundation of Shanxi Province (grant number: 2022JQ-386).

**Institutional Review Board Statement:** Not applicable.

**Informed Consent Statement:** Not applicable.

**Data Availability Statement:** The portion of the data presented in this study are available on request from the corresponding author. The raw data cannot be shared at this time as the data also forms part of an ongoing study.

**Conflicts of Interest:** The authors declare no conflict of interest. The funders had no role in the design of the study; in the collection, analyses, or interpretation of data; in the writing of the manuscript; or in the decision to publish the results.

## Appendix A

In 2014 [18], we adopted a fuzzy clustering algorithm to select TSPs, which was considered mature at the time. With a large batch of thermal error experiments, in 2015 [15], we found that the variability of TSPs and the variability can lead to changes in the collinearity of TSPs, making the model unstable. Furthermore, we also found that the partial regression algorithm is not sensitive to the change of collinearity, so a principal component regression algorithm is proposed to establish a thermal error model, and the stability of the model can be improved. In 2017 [35], we further studied the collinearity and found that the model built by TSPs with high collinearity and the ridge regression algorithm has higher accuracy and stability. At this time, we have been focusing on the improvement of the modeling algorithm, but since then, we have tried many algorithms and found that the improvement effect is not obvious. Until 2020 [10], we found that no matter what algorithm we used, it was impossible to predict all the thermal error data with one model. Furthermore, we believe that the continuous updating of the model is necessary, and the variability of TSPs cannot be ignored. Therefore, we made an in-depth study of the TSPs. Then we found that there is some connection between the correlation and collinearity of TSPs, and the

correlation is very important for the selection of TSPs. In addition, we have studied the stability of the correlation calculation results and published the literature [28]. In the following research, we found that there are two reasons that lead to the problem of TSPs variability: (1) the conflict between correlation and collinearity; (2) the correlation stability. In addition, these are two mathematical problems, that is, the TSPs are not really changed, but the selection method is problematic. Finally, in the research on the thermal error model adaptive update method, this paper systematically studies the reasons for the variability of temperature sensitive points, and proposes a new selection method for temperature sensitive points.

In addition to the research on the selection of TSPs, we simultaneously carried out research on the model update algorithm. We first chose the neural network algorithm because it can learn online. However, the training of a neural network is difficult, because the loss function of a neural network is non-convex, and it is impossible to know in advance whether the training of new data will converge. If some global optimization algorithms are used, such as ant colony and genetic algorithm, it will lead to a significant increase in the amount of calculation and the difficulty of training. In addition, numerous studies on regression models and our experimental results show that the relationship between thermal error and TSPs is dominated by linearity. Therefore, we turn the focus of our research to linear models and propose an update regression algorithm. The update regression algorithm gives the linear model the ability to update online, which is the algorithm used in this study.

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
