# Peer review of "Temperature-Sensitive Point Selection and Thermal Error Model Adaptive Update Method of CNC Machine Tools"

_machines, doi:10.3390/machines10060427_

Round 1

Reviewer 1 Report

The subject of thermal errors introduced by the thermal stress of the machining tool subassembly is topical and interesting. This paper proposed a new adaptive updating method to predict the influence of thermal errors on machining accuracy from less experimental data.

The research is well conducted, but the originality of the research is not accurately stated with respect to published work by the same authors [10], [15], [15], [18], [28], and [33]. The research has no support for physical phenomena occurring during processing. The problem of TSP localization is not clearly formulated. The processing regime and ambient temperature will change the location of the TSP. Heat transfer phenomena are presented by the authors as a mathematical rather than a physical problem. I advise the authors to discuss these issues in the Introduction section.

In section 2.1, details of the sensors used (manufacturer, type, and characteristics) are required. 

How were the sensors placed on the machine tool structure? Is there any indication in the literature regarding the emplacement of the TSPs?

There are two figures 2 and the legend of figure 4 is missing.

What is the error due to replacing the probe with the cutting tool?

Why was the temperature recorded in 10 TSPs if only 2 data were used in the results of the comparison of TSP selection methods? 

The results shown in Table 2 for the two methods did not use measurements from the same TSP.  What is the reason?!  

Validation of the proposed model with results from literature is requested in terms of execution time and prediction accuracy.

Reviewer 2 Report

The paper deals with the thermal error of CNC machine tools that can be reduced by compensation using a thermal error model. The analysis of the selection of temperature-sensitive points (TSPs) in terms of conflict of the collinearity and correlation was presented. Moreover, a new thermal error model adaptive update method was proposed. The new approach adds a regularization term that tends to the old model coefficients in the loss function of the regression algorithm. The paper is suitable for the Machines journal. The topic is well introduced, justified and relevant. However, the weakest element of the paper is the presentation of the results and their discussion. The results presentation is not clear and, in some parts, incomprehensible. The discussion is also not clear and limited. This must be corrected before publication. The language needs correction. The other comments are following:

  • Line 61, 62: “…between model the predicted and…” – something went wrong, syntax.
  • Line 64: “…Zimmermann [13]…” – incorrect reference, should be “…Zimmermann et al. [13]…”. Please check all references.
  • Line 99: “…4machine tool…” – something went wrong.
  • Line 149, 150: “Because of the simple structure and convex loss function of multiple regressing algorithms. They are more stable and easier to train.” – something went wrong, syntax.
  • Line 168-174: this paragraph should be modified. Right now, its structure and style are not acceptable.
  • Line 221: “…can be estimated by Eq. (7).” – this is an awkward introduction of the equation. I suggest modifying this sentence in the following way: “…can be estimated by the following formula/relation/equation:”. Please check and correct (if required) introductions of other equations (e.g., eq. (3), (4), (9), (12), (13), (17), (18), (19), (20)).
  • What is “v” in eq. (6).
  • Not all symbols in eq. (7) and (8) are explained, also in eq. (9), (10), and (11). Please check all equations.
  • Line 238: “…shown in (5). The…” – what is (5)? Do you mean Eq. (5)?
  • Figures on page 9 have incorrect numbers, and the lower one does not have correct captions. Moreover, the space is missing in “Shell(ambient temperature)”.
  • The results shown in Fig. 5 are not discussed and described. What does mean the temperature presented in Fig. 5? How was the thermal error calculated?
  • Captions of Fig. 7, 8, and Table 3: spaces are missing, a capital letter in the bracket.
  • Line 402, 427: the space is missing.
  • There are two equations that have the same number (19).

Reviewer 3 Report

Dear Authors, 

my comments are as follows:

  1. The details of experimental test are not given: material, cutter etc.  What was the wear of the tool and the tool life? One tool was used for 240 minutes?
  2. The details of the temperature measurement are not given. 
  3. There is no validation of the model. 
  4. Figure 5; The temperature stability is observed after ~50 minutes, corresponding to the stability of the dimensions. Why the tests were lasting 240 minutes. It should not be temperature, but the temperature increase. 

Best regards, 

Reviewer

Round 2

Reviewer 1 Report

The detailed explanations provided by the authors to my questions improved the clarity of the paper. I recommend publication of this paper in its current form.

Reviewer 2 Report

The authors improved the paper in-line with my suggestions. Therefore, it may be accepted for publication.

Reviewer 3 Report

Dear Authors, 

I accept your modifications and answers. 

reviewer